# SCALING PROPERTIES OF DIFFUSION MODELS FOR PERCEPTUAL TASKS

## ABSTRACT

In this paper, we argue that iterative computation, as exemplified by diffusion models, offers a powerful paradigm for not only image generation but also for visual perception tasks. First, we unify few of the mid-level vision tasks as image to image translations tasks ranging from depth estimation to optical flow to segmentation. Then, through extensive experiments across these tasks, we demonstrate how diffusion models scale with increased compute during both training and inference. Notably, we train various dense and Mixture of Expert models up to 2.8 billion parameters, and we utilize increased sampling steps, use various ensembling methods to increase compute at test time. Our work provides compelling evidence for the benefits of scaling compute at train and test time for diffusion models for visual perception, and by studying the scaling properties carefully, we were able to archive same performance of the state-of-the-art with less compute.

## 1 INTRODUCTION

Recently, Diffusion models and Auto Regressive models have emerged as a powerful technique for generating images and videos. With some loss of generalizations Diffusion models can be viewed as auto regressive models in frequency space (Dieleman, 2024). These models have shown excellent scaling behaviours for image and video generation tasks. One could attribute part of this success to iterative computation, which at inference the model spends more compute, usually about one or two orders of magnitude compared to single forward pass.

While the success story of Diffusion and Auto Regressive models on generative tasks is unprecedented, in this work we ask, can these iterative prediction models be used for perceptual tasks (inverse problems) and leverage the benefits of scaling test time computation. Marigold (Ke et al., 2024) and FlowDiffuser (Luo et al., 2024) partially answered this question for depth estimation and optical flow *individually*, and showed that diffusion models are indeed suited these perceptual tasks. Our work explores a range of perceptual tasks from low-level optical flow to mid-level depth estimation and more complex semantic segmentation and occlusion reasoning, under a unified framework, and we carefully study the compute scaling behaviours at train and test time.

To study the scaling properties of diffusion models for inverse problems both at training and test-time, we pre-train various dense models sizes from 14 million to 1.8 billion parameters, and up to 2.8 billion parameter mixture of experts models for the class-conditional image generation task. We fine-tune these pre-trained models on various downstream tasks. We study the scaling properties at training by varying model size, data resolution, and pre-training compute. In addition to this, we apply efficient training strategies such as upcycling dense checkpoints to mixture-of-experts models without training them from scratch. At inference, we evaluate our models on downstream tasks, with various test-time compute allocation techniques. These include scaling number of diffusion steps, test-time ensembling, and increasing number of model experts.

In summary, this paper argues in favour of iterative feedback computation for visual perception tasks, presenting three key findings. Through extensive ablation studies, we explore various methods to scale computational resources during training and inference. Our results demonstrate that by utilizing these scaling laws, we can achieve competitive performance across a diverse range of perception tasks, from low-level optical flow to complex amodal segmentation. Furthermore, we train a unified model architecture that employs expert routing, enabling it to effectively address multiple perception tasks within a single model.

Figure 1: **A Unified Framework:** We fine-tune a pre-trained Diffusion Model, for visual perception tasks. We take a RGB image, and a conditional image (i.e. next video frame, occlusion mask, etc.), along with the noised image of the ground truth prediction. Our model generates predictions for various visual tasks such as depth estimation, optical flow prediction, and amodal segmentation, based on the conditional task embedding.

## 2 RELATED WORK

**Biological Inspirations:** Visual information processing in both biological systems involves complex interconnections. In human visual cortices, feedforward connections link low to high-level areas in dorsal and ventral pathways, while feedback connections allow for refined processing over time (Felleman & Van Essen, 1991; Lamme & Roelfsema, 2000; Kravitz et al., 2013). For example, in inferotemporal cortex for the neurons that are selective for face, and their first response for a face stimuli are simple classifier of where the stimuli is a face or not, but with over time, the cells responses will contain expressions and identity (Lamme & Roelfsema, 2000).

**Generative Modeling:** Generative modeling has been studied under various methods, VAEs (Kingma, 2013), GANs (Goodfellow et al., 2014), Normalizing Flows (Rezende & Mohamed, 2015), Auto-Regressive models (van den Oord et al., 2016), and Diffusion models (Sohl-Dickstein et al., 2015; Ho et al., 2020). Among these, Denoising Diffusion Probabilistic Models (DDPMs) (Ho et al., 2020) have shown impressive scaling behaviors and have become a de facto generative tool for many image and video generation models. Notable examples include Latent Diffusion Models (Rombach et al., 2022) which enhanced efficiency by operating in a compressed latent space, Imagen (Saharia et al., 2022) which generates samples in pixel space with increasing resolution and Consistency Models (Song et al., 2023) which aim to accelerate sampling while maintaining generation quality. Apart from diffusion based models, Parti (Yu et al., 2022) and MARS (He et al., 2024) showcased the potential of auto regressive models image generation tasks and the Muse architecture (Chang et al., 2023) introduced a masked image generation approach using transformers.

**Scaling Diffusion Models:** Diffusion modeling has shown impressive scaling behaviors in terms of data, model size, and compute. Latent Diffusion Models (Rombach et al., 2022) first showed that, scaling the training with large-scale web datasets and compute can archive high quality image generation results with U-Net model. DiT (Peebles & Xie, 2023) studied the scaling behavior of diffusion models with transformer architecture and showed that, transformer models have good scaling for class conditional image generation. Later, *Li et al.*(Li et al., 2024) studied the scaling laws of text-to-image diffusion models for alignment. Recently, *Fei et al.*(Fei et al., 2024a) trained DiT models up to 16 billion parameters with mixture of experts and achieved high-quality image generation results. Finally, another way to scale transformer models is via upcycling. *Komatsuzaki et al.* (Komatsuzaki et al., 2022) used sparse upcycling to learn a mixture of experts model from a dense transformer model without needing to pretrain a mixture of experts model.

**Diffusion Models for Perception Tasks:** While diffusion models have an impressive track record of generating images and videos, they have also been used for various downstream visual tasks. For example, diffusion models have been used for estimating depth (Ji et al., 2023; Duan et al., 2023; Saxena et al., 2023; 2024; Zhao et al., 2023), and recently Marigold (Ke et al., 2024) and GeoWizard (Fu et al., 2024) showed impressive results by repurposing pretrained diffusion models for monocular depth estimation. Diffusion models with few modifications are used for semantic segmentation for categorical distributions (Hoogeboom et al., 2021; Brempong et al., 2022; Tan et al., 2022; Amit et al., 2021; Baranchuk et al., 2021; Wolleb et al., 2022), instance segmentation (Gu et al., 2024), and panoptic segmentation (Chen et al., 2023). Additionally, diffusion models are also used for optical flow estimation (Luo et al., 2024; Saxena et al., 2024) and 3D understanding (Liu et al., 2023; Jain et al., 2022; Poole et al., 2022; Wang et al., 2023; Watson et al., 2022).

## 3 GENERATIVE PRE-TRAINING

Our approach involves first pre-training diffusion models for conditional image generation task, and we utilize a diffusion transformer (DiT) backbone. For the pre-training we follow DiT recipes (Peebles & Xie, 2023).

Starting with a target image $I \in \mathbb{R}^{u \times u \times 3}$ as an RGB image, where the resolution of the image is $u \times u$, our pretrained, frozen Stable Diffusion variational autoencoder (Rombach et al., 2022) compresses the target to a latent $z_0 \in \mathbb{R}^{w \times w \times 4}$, where $w = u/8$. Gaussian noise is added at sampled time steps to obtain a noisy target latent and noisy samples are generated as:

$$z_t = \sqrt{\alpha_t} \cdot z_0 + \sqrt{1 - \alpha_t} \cdot \epsilon_t \tag{1}$$

for timestep $t$. The noise is distributed as $\epsilon \sim \mathcal{N}(0, I)$, $t \sim \mathrm{Uniform}(T)$, with $T = 1000$ and $\alpha_t := \prod_{s=1}^{t}(1 - \beta_s)$, with $\{\beta_1, \ldots, \beta_T\}$ as the variance schedule of a process.

In the denoising process, the class-conditional DiT $f_\theta(\cdot)$ parameterized by learned parameters $\theta$ gradually removes noise from $z_t$ to obtain $z_{t-1}$. Parameters $\theta$ are updated by noising $z_0$ with sampled noise $\epsilon$ at a random timestep $t$, and computing the noise estimate

$$\theta^* = \arg\min_\theta \mathcal{L}_\theta(z_t, \epsilon_i) = \arg\min_\theta \frac{1}{n} \sum_{i=1}^{n} (\epsilon_i - \hat{\epsilon}_i)^2 \tag{2}$$

as a mean squared loss applied between the generated noise and noise estimated by the $\theta$ parameterized DiT, in an $n$ batch size sample.

### 3.1 MODEL SIZE

We pre-train six different dense DiT models as in Table 1, increasing model size by varying the number of layers and hidden dimension size. We follow the pre-training recipe in DiT (Peebles & Xie, 2023), using Imagenet-1K (Russakovsky et al., 2015) as our pretraining dataset with the same amount of total training iterations. we trained all the models for 400k iterations with a fixed learning rate of $1e-4$ for all the models and a batch size of 256. Fig 2 shows that larger models converge to lower loss with a clear power law behavior. We show the train loss as a function of compute (in MACs), and out predictions show a power law relationships of $L(C) = 0.23 \times C^{-0.0098}$.

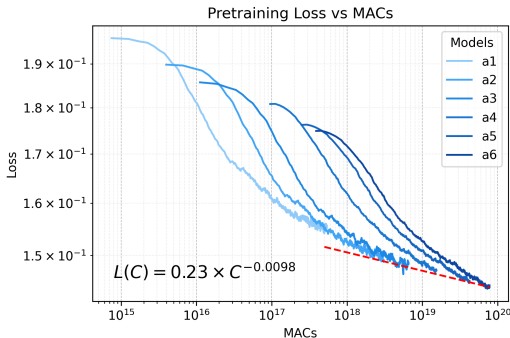

Figure 2: **Scaling at Model Size:** For generative pre-training of DiT models we see a clear scaling behaviors as we increase the model size.

### 3.2 MIXTURE OF EXPERTS

We pre-train Sparse Mixture of Experts (MoE) models (Shazeer et al., 2017), following the model configurations in (Fei et al., 2024b) for S/2 and L/2 configurations. We use three different MoE configurations listed in table 2, scaling the total parameter count by increasing hidden size, number of experts, layers, and attention heads. Each MoE block activates the top-2 experts per token and has a shared expert that is used by all tokens. To alleviate issues with expert balance, we use the proposed expert balance loss function from (Fei et al., 2024b) to distribute the load across experts more efficiently. Sparse MoE pre-training allows for a higher parameter count while increasing throughput, making it more compute efficient than training a dense DiT model. We train our DiT-MoE models with the same training recipe as the dense DiT models, using ImageNet-1K as the pre-training dataset and training for 400K iterations with batch size 256.

| Model | Params | Dimension | Heads | Layers |
|-------|--------|-----------|-------|--------|
| a1 | 14.8M | 256 | 16 | 12 |
| a2 | 77.2M | 512 | 16 | 16 |
| a3 | 215M | 768 | 16 | 20 |
| a4 | 458M | 1024 | 16 | 24 |
| a5 | 1.2B | 1536 | 16 | 28 |
| a6 | 1.9B | 1792 | 16 | 32 |

| Model | Active / Total | Dim | Heads | Layers |
|-------|----------------|-----|-------|--------|
| S/2-8E2A | 71M / 199M | 384 | 6 | 12 |
| S/2-16E2A | 71M / 369M | 384 | 6 | 12 |
| L/2-8E2A | 1.0B / 2.8B | 1024 | 16 | 24 |

Table 1: **Dense DiT Models:** We scale dense DiT model size by increasing hidden dimension and number of layers linearly while keeping number of heads constant following (Yang et al., 2022; Touvron et al., 2023).

Table 2: **MoE DiT Models:** We scale the MoE DiT models by increasing dimension size, number attention heads, layers, and experts following (Fei et al., 2024b).

## 4 FINE-TUNING FOR PERCEPTUAL TASKS

During fine tuning, we utilize the image-to-image diffusion process from (Ke et al., 2024) and (Brooks et al., 2023) as our training recipe, and we pose all our visual tasks as conditional denoising diffusion generation. Give an RGB image $I \in \mathbb{R}^{u \times u \times 3}$ and it's pair ground truth image $D \in \mathbb{R}^{u \times u \times 3}$, we first project them to latent space, $i_0 \in \mathbb{R}^{w \times w \times 4}$ and $d_0 \in \mathbb{R}^{w \times w \times 4}$ respectively. We only add noise the ground truth latent to get $d_t$ and concatenate it with the RGB latent to get a tensor of $z_t = \{i_0, d_t\}$. The first convolution layer of the DiT model is modified to match the doubled number of channels in the input, and it's values are reduced by half to make sure the predictions are the same if the inputs are just RGB images (Ke et al., 2024). Finally, we simply perform diffusion training by denoising the ground truth image.

This approach allow us to unify all the visual tasks as image-to-image translation. We ablate various fine tuning compute scaling behaviors on the monocular depth estimation task and report absolute error and delta1 accuracy. We use the best configurations from the depth estimation ablation study to fine-tune for other visual tasks.

### 4.1 EFFECT OF MODEL SIZE

We fine-tune the pre-trained a1-a6 dense models on the depth estimation task to study the effect of model size. We scale model size as shown in Table 1, increasing total layers and hidden size. We follow the pre-training recipe of the original DiT (Peebles & Xie, 2023), using ImageNet-1K (Russakovsky et al., 2015) as our pre-training dataset with the same amount of total training iterations. Fig 3 shows that larger dense DiT models converge to lower fine-tuning loss, presenting a clear power law scaling behavior. We show the train loss as a function of compute (in MACs), and our model predictions show a power law relationship in both depth Absolute Relative error and depth Delta1 error.

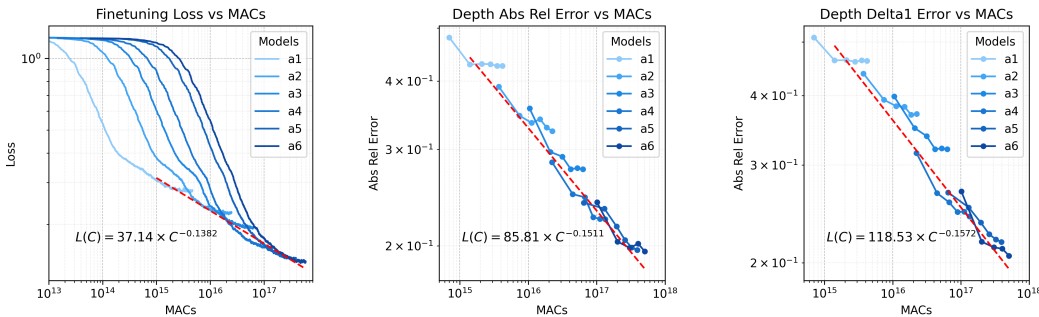

Figure 3: **Effect of Model Size:** We fine-tune a1-a6 models on the Hypersim dataset for 30K iterations with a exponential decay learning rate schedule from $3e-5$ to $3e-7$. We observe a strong correlation between the fine-tuning scaling law and validation metric scaling laws.

## 4.2 EFFECT OF PRE-TRAINING COMPUTE

We also investigate the behavior of the fine-tuning process as we scale the number of pre-training steps for the DiT backbone. We train the A4 model with a varied number of pre-training steps while keeping all other hyperparameters constant. We then fine-tune these four models on the same depth estimation dataset. Fig 4 displays the power law scaling behavior of the validation metrics for depth estimation as we increase DiT pre-training steps.

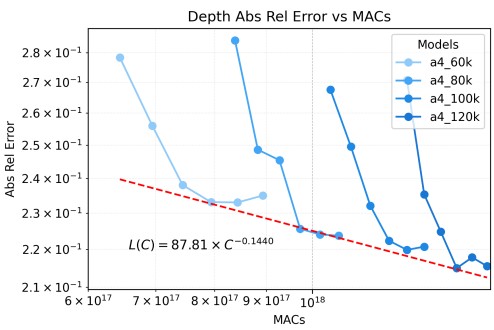 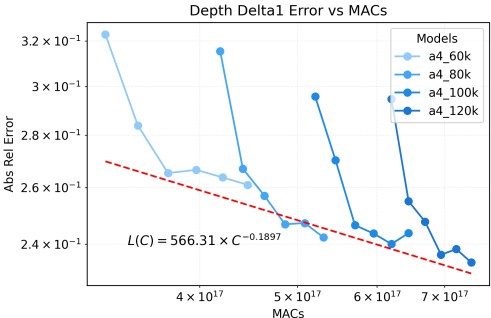

Figure 4: **Effect of scaling model pre-training compute on depth estimation:** (a) Depth Absolute Relative Error vs. MACs. (b) Depth Delta1 Error vs. MACs. We pre-train four A4 models with 60K, 80K, 100K, and 120K steps with a batch size of 1024 at a fixed learning rate of $1e-4$. These models are then fine-tuned for 30K steps on the Hypersim depth estimation dataset with a batch size of 32. We observe a clear power law as we increase the DiT pre-training compute across depth estimation validation metrics.

## 4.3 EFFECT OF IMAGE RESOLUTION

The total number of tokens per image also affects the total compute spent during training. For each forward pass, we can scale the number of FLOPs used by simply scaling up the resolution of the image, which will increase the number of tokens used to represent the image embedding. By increasing the resolution and the number of tokens, we can increase the amount of information the model can learn from at training time to build stronger internal representations, which can in turn improve downstream performance. We use dense DiT-XL models with resolutions of 256x256 and 512x512 from (Peebles & Xie, 2023) and we pre-train DiT L/2-8E2A models with 256x256 and 512x512 resolutions following the recipe in (Fei et al., 2024b). We then fine-tune each of these models with the corresponding resolution for the depth estimation task. In Fig 5, we present scaling laws for image resolution during fine-tuning on depth estimation.

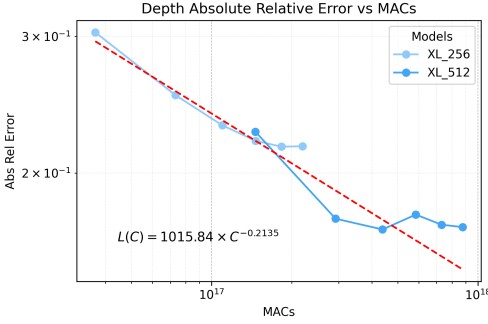 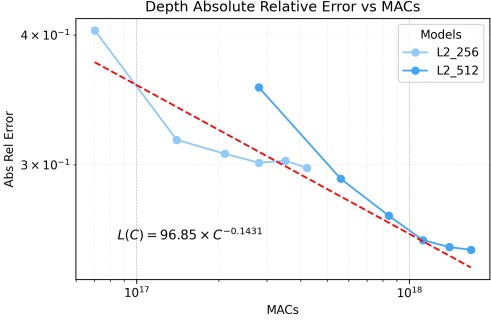

Figure 5: **Effect of Image Resolution.** We fine-tune DiT-XL and DiT-MoE L/2 models with resolutions of 256x256 and 512x512. We observe a power law when increasing image resolution during training. By scaling the number of tokens per image by 4X, we achieve strong performance on Depth Absolute Error, displaying the effect of increasing total dataset tokens for dense visual perception tasks such as depth estimation.

### 4.4 Effect of Upcycling

Sparse MoE models are efficient options for increasing the capacity of a model, but pre-training an MoE model from scratch can be expensive. One way to alleviate this issue is Sparse MoE Upcycling (Komatsuzaki et al., 2023). Upcycling converts a dense transformer checkpoint to an MoE model by copying the MLP layer in each transformer block $E$ times, where $E$ is the number of experts, and adding a learnable router module to send each token to the top-$k$ selected experts. The outputs of the selected experts are then combined in a weighted sum at the end of each MoE block.

In Fig 6, we show the effect of up-cycling various dense models after they have been fine-tuned for depth estimation. We continue the fine-tuning on each up-cycled model, which also tunes the randomly initialized router module weights. Fig 6 displays the scaling laws for up-cycling, providing an average improvement of 5.3% on Absolute Relative Error and 8.6% on Delta1 error.

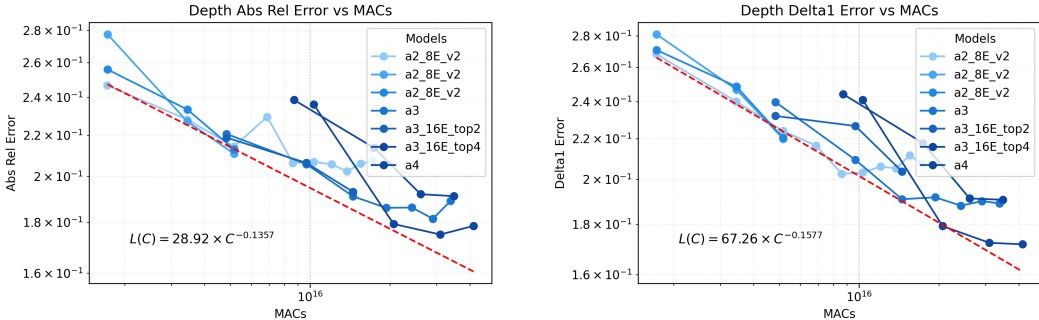

Figure 6: **Effect of Upcycling.** We upcycle A2, A3, and A4 models for a varying number of experts as shown in the figure. We continue fine-tuning each upcycled model for another 15K iterations at a batch size of 128 on the Hypersim depth estimation dataset. We observe a clear scaling law between using more MACs with upcycling and decreasing Absolute Relative and Delta1 error rates. We also observe that upcycling can achieve equivalent or superior performance to our dense A5 and A6 checkpoints, each of which utilize more compute during pre-training and fine-tuning. We also note that increasing the total number of experts and the total active experts improves the downstream performance.

## 5 Scaling Test-Time Compute

Scaling inference compute has been applied for autoregressive Large Language Models (LLMs) to improve performance on long-horizon and complex reasoning tasks (Brown et al., 2024; Snell et al., 2024; El-Refai et al.). In this section, we explore methods for scaling test-time compute for perceptual tasks with diffusion models.

We present the general inference pipeline in Fig 7. We use the original Stable-Diffusion VAE to encode the input image into the latent space (Rombach et al., 2022). Then, we sample a target noise latent from a standard Gaussian distribution, which will be iteratively denoised to generate the downstream prediction using the same noise schedule as fine-tuning. We apply the non-Markovian sampling with re-spaced steps to speed up inference as proposed in DDIM (Song et al., 2021). The target latent is then decoded by the VAE to obtain the final downstream task prediction. For the depth estimation downstream task, we follow the procedure in Marigold by averaging the final decoded prediction across the channel dimension (Ke et al., 2024). In Fig 7, we summarize our three approaches to scaling test-time compute for diffusion.

### 5.1 Effect of Inference Steps

One natural way of scaling diffusion inference is by increasing denoising steps. Since the model is trained to denoise the input at various timesteps, we can scale the number of diffusion denoising steps at test-time to produce more accurate predictions. This paradigm is also reflected in the generative case, where the corruption process of diffusion pushes the model to learn a coarse-to-fine denoising

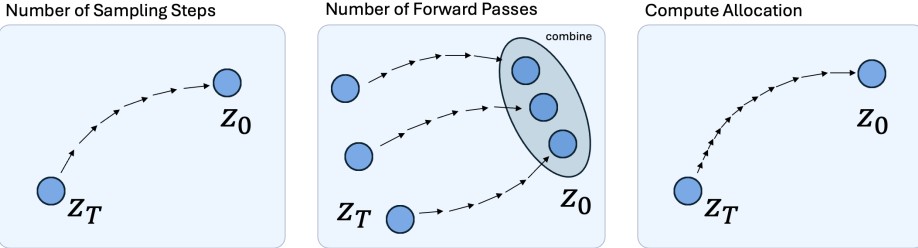

Figure 7: **Inference Scaling:** Diffusion models by design allows scaling of test time compute natively. First, we can simply increase the number of denoising steps to increase the compute spent at inference. Second, since we are estimating deterministic outputs, we can initialize the noise multiple time and combine the predictions to get a better estimate the output. Finally, we can also allocate different compute budget for low and high frequency denoising with cosine schedule.

trajectory. We can exploit this paradigm for the discriminative case by increasing the number of denoising steps, which will generate finer predictions. In Fig 8, we observe that increasing the total test-time compute by simply increasing the number of diffusion sampling steps provides substantial gains in depth estimation performance.

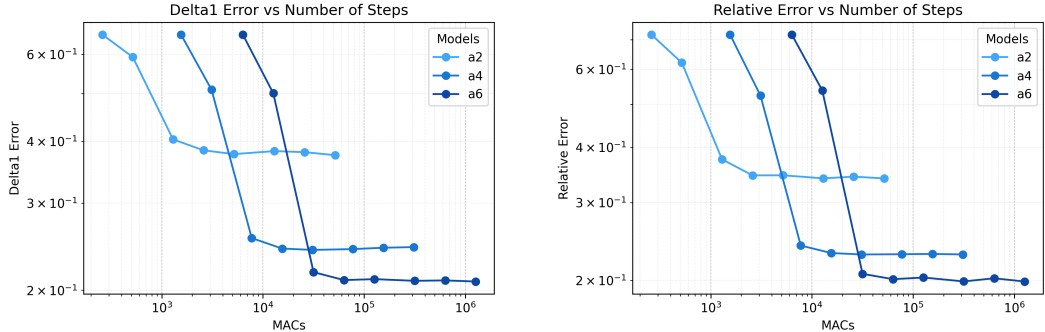

Figure 8: **Effect of Number of Sampling Steps.** (a) Delta1 Error vs. Number of Steps. (b) Absolute Relative Error vs. Number of Steps. We show the effect of scaling test-time compute by increasing the number of diffusion sampling steps. For each model, we sample for $T \in [1, 2, 5, 10, 20, 50, 100]$ steps with the DDIM sampler. We show a clear power law scaling behavior in (a) and (b), displaying the effectiveness of simply utilizing the iterative nature of diffusion to improve downstream performance.

### 5.2 EFFECT OF TEST TIME ENSEMBLING

We also explore scaling inference compute through test-time ensembling. Here, we take advantage of the fact that denoising different noise latent initializations will generate different results. In test-time ensembling, we compute $N$ forward passes for each input sample, and reduce the outputs through one of two methods. The first technique is naive ensembling: simply compute $N$ forward passes and use the pixel-wise median or mean across all outputs as the prediction. This is the median/mean aggregation technique, which provides a straightforward method for combining predictions across multiple noise initializations efficiently. The second ensembling technique presented in (Ke et al., 2024) is median compilation, where we collect predictions $\{\hat{d}_1, \ldots, \hat{d}_N\}$ that are affine-invariant, jointly estimate scale and shift parameters $\hat{s}_i$ and $\hat{t}_i$, and minimize the distances between each pair of scaled and shifted predictions $(\hat{d}'_i, \hat{d}'_j)$ where $\hat{d}' = \hat{d} \times \hat{s} + \hat{t}$. For each optimization step, we take the pixel-wise median $\boldsymbol{m}(x, y) = \text{median}(\boldsymbol{d}'_1(\hat{x}, y), \ldots, \boldsymbol{d}'_N(\hat{x}, y))$ to compute the merged depth $\boldsymbol{m}$. This iterative optimization on spatial alignment paired with extra regularization is used to produce the final depth prediction. Since it requires no additional ground truth, we can scale this ensembling technique by increasing $N$ to utilize more test-time compute.

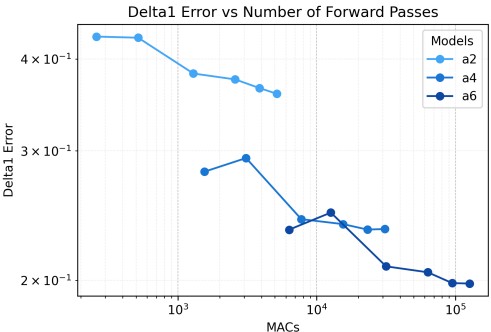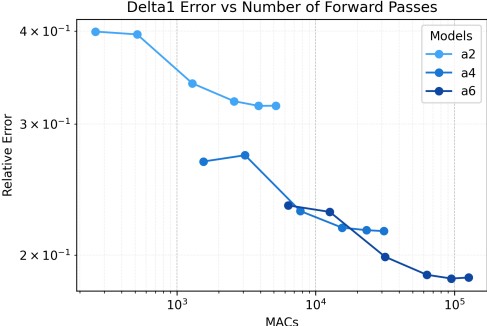

Figure 9: **Effect of Test Time Ensembling.** (a) Delta1 Error vs. Number of Forward Passes. (b) Absolute Relative Error vs. Number of Forward Passes. We observe that using an ensembling strategy by merging multiple predictions from distinct noise initializations displays power law scaling behavior. Here, we increase test-time compute by increasing the number of forward passes at each denoising step by denoising $N$ noise latents. We apply this method using values of $N \in [1, 2, 5, 10, 15, 20]$.

### 5.3 EFFECT OF NOISE VARIANCE SCHEDULE

In diffusion noise schedulers, we can also define a schedule for the variance of the Gaussian noise applied to the image over the total diffusion timesteps $T$. This schedule determines the noise level applied to the image at each step $t$. Tuning the noise variance schedule allows for the reorganization of compute by providing the option to spend more FLOPs on denoising steps early or later in the noise schedule. We experiment with using three different noise level settings for DDIM: linear, scaled linear, and cosine. Cosine scheduling from (Nichol & Dhariwal, 2021) linearly declines from the middle of the corruption process, allowing for a more balanced noise schedule that doesn't corrupt the image too quickly as in linear schedules. In Fig 10, we observe that the cosine noise variance schedule outperforms the default linear schedule for DDIM on the depth estimation task.

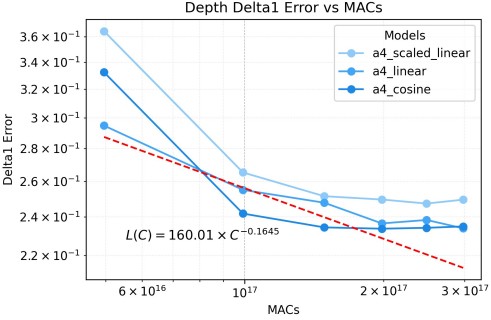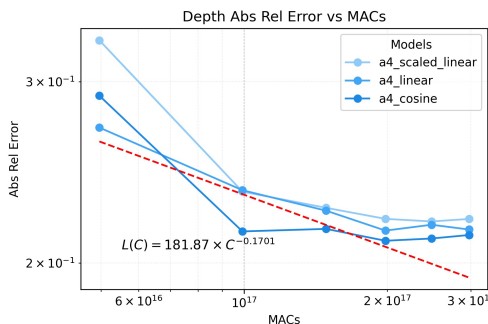

Figure 10: **Effect of Noise Variance (Beta) Schedule.** We fine-tune A4 models with three different beta schedules: linear, scaled linear, cosine. We observe a power law when using different noise variance schedules. Reallocating compute with the cosine schedule to spend more time denoising at earlier timesteps by slowly adding noise provides improved Delta1 and Absolute Relative Error rates.

## 6 PUTTING IT ALL TOGETHER

Using the lessons from our experiments on depth estimation, we train diffusion models for optical flow prediction and amodal segmentation. Compared to prior methods, we utilize much smaller models, less training compute, and smaller data resolution, while achieving similar results to state-of-the-art for the respective tasks. We show that using diffusion models while considering efficient methods to scale training and test-time compute can provide substantial performance benefits on visual perception tasks. Finally, we train a unified model, capable of performing all three visual

perception tasks previously mentioned, displaying the generalizability of our method. See Fig 11 for to see the quality of our predicted samples.

## 6.1 DEPTH ESTIMATION

We combine our findings from the ablation studies on depth estimation to create a model with the best training and inference configuration. We train a DiT-XL model from (Peebles & Xie, 2023) on depth estimation data from Hypersim for 30K steps with a batch size of 1024, resolution of 512x512, and a learning rate exponentially decaying from $1.2e-4$ to $1.2e-6$. We use the median compilation strategy at test-time with a cosine noise variance schedule. As shown in Table 3, our model is able to achieve slightly improved validation performance over Marigold on the Hypersim dataset, while being trained on a lower resolution and less pre-training data.

| Metric | Model | Resolution | Value |
|---|---|---|---|
| Delta 1 Accuracy | DiT-XL/2 | 512x512 | 0.876 |
| Abs Relative Error | DiT-XL/2 | 512x512 | 0.136 |
| Delta 1 Accuracy | Marigold | 480x640 | 0.8754 |
| Abs Relative Error | Marigold | 480x640 | 0.135 |

Table 3: **Depth Estimation Comparison on Hypersim.** We achieve almost equivalent performance on depth estimation over Marigold. We utilize a smaller resolution and a smaller DiT-XL/2 model that has been trained with less data and parameters than the Stable Diffusion model used in Marigold. These results display the effectiveness of applying our scaling techniques on depth estimation.

## 6.2 OPTICAL FLOW PREDICTION

We use a similar configuration as the depth estimation model for optical flow training. We train a DiT-XL model on the FlyingChairs dataset for 4K steps with batch size of 1024, resolution of 512x512, and learning rate exponentially decaying from $1.2e-4$ to $1.2e-6$. We compare our model's performance with other specialized optical flow prediction techniques in Table 4.

| Method | Chairs test |
|---|---|
| DeepFlow | 3.53 |
| FlowNetS | 2.71 |
| FlowNetS+v | 2.86 |
| FlowNetS+ft | 3.04 |
| FlowNetS+ft+v | 3.03 |
| FlowNetC | 2.19 |
| FlowNetC+v | 2.61 |
| FlowNetC+ft | 2.27 |
| FlowNetC+ft+v | 2.67 |
| **Ours** | 5.826 |

Table 4: **Comparison with Specialized Techniques.** We evaluate our optical flow model on the FlyingChairs validation set. We observe our model is able to achieve similar results compared to specialized methods such as DeepFlow (Weinzaepfel et al., 2013) and FlowNet (Fischer et al., 2015) in terms of end-point error. Our model is trained on a much smaller dataset compared to the specialied FlowNet method, which is trained on a variety of optical flow datasets.

## 6.3 AMODAL SEGMENTATION

For amodal segmentation, we further scale up our fine-tuning approach. We continue to train a DiT-XL model on the pix2gestalt dataset (Ozguroglu et al., 2024) for 6K steps with a batch size of 4096, resolution of 256x256, and learning rate exponentially decaying from $1.2e-4$ to $1.2e-6$.

Figure 11: **Amodal Segmentation and Depth Estimation Examples:** On the left, we show the results from our amodal segmentation models, where the model sees RGB image, and segmentation of the occluded object. The task is to predict the amodal image, and out predictions are very similar to the ground truth labels. On the right, we show predictions from our depth estiomation model, with rgb, gtound truth and predction on the first, second and thrid columns respectively.

| Method | IOU |
|---|---|
| pix2gestalt | 0.88 |
| **Ours** | 0.87 |

Table 5: **Comparison with pix2gestalt**. Our model is able to achieve almost equivalent IOU performance as pix2gestalt (Ozguroglu et al., 2024) on their validation set. Our model was pre-trained on only ImageNet-1K, whereas pix2gestalt uses Stable Diffusion, which is trained with at least an order of magnitude more data at a higher resolution and more training compute.

## 6.4 ONE MODEL FOR ALL

Finally, we train a unified DiT-XL model for each of the different tasks. We train this model on the mixed dataset for 10K steps with a batch size of 1024, resolution of 512x512, and learning rate exponentially decaying from $1.2e-4$ to $1.2e-6$. To train this generalist model, we modify the DiT-XL architecture by replacing the patch embedding layer with a separate `PatchEmbedRouter` module, which routes each VAE embedding to a specific input convolutional layer based on the dataset from which we sample the VAE embedding. This ensures the DiT-XL model is able to distinguish between the task-specific embeddings during fine-tuning.

## 7 CONCLUSION

In our work, we examine the scaling properties of diffusion models for visual perception tasks. We explore various approaches to scale training, including increasing model size, mixture-of-experts models, increasing image resolution, and upcycling. We also realize the effectiveness of scaling test-time compute by exploiting the iterative nature of diffusion to reallocate compute to boost downstream performance. Our experiments provide strong evidence of scaling with power laws across various training and inference scaling techniques. We hope to inspire future work in scaling training and test-time compute for iterative generative paradigms such as diffusion for perception tasks.

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
