# OpenReview forum: "Scaling Diffusion Models for Downstream Prediction"
_ICLR.cc/2025/Conference — ICLR 2025 Conference Withdrawn Submission_

### Official Review · Reviewer_u13o · 2024-10-26

**Soundness:** 2
**Presentation:** 2
**Contribution:** 1
**Rating:** 3
**Confidence:** 4

**Summary:**

This paper shows how to scale up diffusion models for downstream visual perception tasks.

**Strengths:**

The motivation for this paper is meaningful, given that most existing methods focus on fine-tuning Stable Diffusion 2.1. Thus, exploring the scaling laws for fine-tuning diffusion models in visual perception tasks is an important area of inquiry.

**Weaknesses:**

1. The conclusions and findings of this paper are ambiguous, making it challenging to derive useful insights from the content.

2. The results presented in this paper lack credibility, as there is no fair comparison with existing models, such as Marigold and GeoWizard, particularly in the context of depth estimation.

3. There are several alternative fine-tuning paradigms that do not adhere to the diffusion protocol, such as flow matching fine-tuning and deterministic fine-tuning. It is crucial to discuss these paradigms in this work, as exploring scaling laws involves understanding how different fine-tuning approaches can significantly impact efficiency and performance.

4. Given that this work only pre-trains the DIT series on ImageNet, I am not confident in the generalization performance of these models on unseen scenes.

**Questions:**

1. how do you design the architecture for optical flow task, it lacks detailed explanation in the paper.
2. the efficiency and performance of the mixture-of-experts should be discussed, and compared with the results of direct finetuning from models like Stable Diffusion 2 and Stable Diffusion XL.

**Details Of Ethics Concerns:**

no ehtics review needed

---

### Official Review · Reviewer_n5cn · 2024-10-27

**Soundness:** 2
**Presentation:** 3
**Contribution:** 2
**Rating:** 3
**Confidence:** 5

**Summary:**

The paper explores the scaling behaviors of diffusion models when applied to visual perception tasks like depth estimation, optical flow prediction, and semantic segmentation. The authors investigate the convergence behavior concerning the scaling of various aspects of the diffusion models, and propose a unified model addressing the three tasks above. Extensive experiments are made for ablation study, thus providing a reference for future model structure design.

**Strengths:**

1. Through detailed ablation studies, the paper provides a rigorous exploration of how model size, pre-training compute, image resolution, and test-time compute affect performance.
2. A key insight is the identification of power-law scaling across various metrics (loss, delta1 accuracy) as compute resources increase, showing that the performance continues to improve predictably with more computing.

**Weaknesses:**

1. Figure 3, Figure 4, Figure 5, Figure 8, Figure 9 incorrectly labeled either y or x-axis. Figure 6’s legend is incorrect. Figure 6’s caption compares the result of upcycling with a5 and a6, which are not even included in the plot.
2. While the experiments are comprehensive, limited datasets are used for perceptual tasks. All the conclusions of the scaling behavior are empirical, so the generalization of conclusions is questionable due to inadequate datasets applied. It is highly recommended to extend the study to more diverse and complicated datasets.
3. No experiments about the unified model are included, but that is supposed to be the largest innovation in this work. The presented experiments on perceptual tasks are still individual models.
4. The results of the scaling behavior follow the “the more resources, the better results” intuition. This, however, does not clear the bar for novelty without additional insights.

**Questions:**

1. What are the “three key findings” (line 49) mentioned in the introduction part?
2. How is the power law approximated? It is a bit inconsistent with the depth absolute relative error and the depth delta1 error. Does this indicate a potential inability for generalization?
3. Could you please explain more about the unified model, including its task embedding, performance on each task compared with SOTA, and computation complexity?

---

### Official Review · Reviewer_uVTV · 2024-11-01

**Soundness:** 3
**Presentation:** 2
**Contribution:** 2
**Rating:** 5
**Confidence:** 4

**Summary:**

In this paper, the authors point out that iterative computation methods like diffusion models are not only effective for image generation but also suitable for perceptual downstream tasks. They propose a unified framework to validate their viewpoint and conduct extensive experiments to demonstrate the scalability performance during training and inference (training: dense, MoE, testing: increasing the number of sampling steps, using various ensembling methods). They prove the effectiveness of scaling up and utilize these findings to achieve comparable performance to existing methods with fewer computational resources.

**Strengths:**

1.The paper proposes a unified framework that treats various mid-level vision tasks as image-to-image translation tasks
2.Through extensive experiments, the paper demonstrates how diffusion models scale during both training and inference with increased computational resources, providing insights into the model's potential and limitations.
3.The authors also train a unified model architecture that employs expert routing, enabling it to effectively address multiple perception tasks within a single model

**Weaknesses:**

1.The writing of this paper is fragmented, focusing too much on technical details and experimental results, oversee the connections between each section, without adequately discussing the limitations of the model，making it read like an experimental report.
2.The unified framework has been explored by former works[1],so the techinal contribution of this paper is limited.
3.The experiments of downstream perception task is not enough, the authors should compare with more methods to demonstrate their superiority
[1]Lee H Y, Tseng H Y, Yang M H. Exploiting Diffusion Prior for Generalizable Dense Prediction[J]. arXiv preprint arXiv:2311.18832, 2023.

**Questions:**

Generally, I appreciate the authors to do many experiments to show the scailing law of diffusion models for downstream tasks,but I think there are still many points to be improved. I have the following questions:
1. The authors should do more experiments to show their superiority over other methods .
2. I'm not quite sure what the authors' evaluation metrics are, for example, the 'value' in depth estimation，I think the authors should clarify it.

---

### Official Review · Reviewer_4dMJ · 2024-11-03

**Soundness:** 2
**Presentation:** 1
**Contribution:** 2
**Rating:** 3
**Confidence:** 4

**Summary:**

- This paper considers to exploit diffusion models for various perceptual tasks, for example, depth estimation, optical flow, and segmentation.

- Diffusion models of various sizes have been trained, along with MoE architectures, to investigate the scaling properties of diffusion models.

- At inference time, the authors have explored different strategies for ensembling the predictions.

**Strengths:**

In this paper, the authors have conducted extensive experiments on training models of various sizes, which shows,

(1) at the stage of pre-training for image generation, larger model demonstrate converge to lower loss with a clear power law behaviour;

(2) at the stage of downstream fine-tuning for perceptual tasks, larger dense DiT models converge to lower fine-tuning loss, also presenting a clear power law scaling behaviour;

(3) longer pre-training will present better performance for downstream perceptual tasks.

(4) larger resolution can be beneficial for downstream perceptual tasks

(5) Scaling test-time computation is beneficial.

**Weaknesses:**

- The paper reads like a technical report, rather than a scientific paper, as it does not provide much insights.

- The writing is not clear enough, for example, there is no details provided for training optical flow and modal segmentations, what are the data format, and there is no supplementary material to explain all these.

- The experiments on optical flow and segmentation are not convincing, for example, the baseline methods are very old, and the benchmark test set is artificial, it should at least to compare with RAFT, on Sintel or something similar.

- The performance on optical flow is not satisfactory, though the authors claim `the model achieve similar results to specialised methods', it is clear that the gap is large .....

- On amodal segmentation, the authors should also evaluate on  Occluded and Separated COCO, as pix2gestalt did.

- There has been many work exploring the discriminative learning with diffusion model,  the authors have missed them in reference:

[1] DatasetDM: Synthesizing Data with Perception Annotations Using Diffusion Models, In NeurIPS 2023

[2] Open-vocabulary Object Segmentation with Diffusion Models. In ICCV 2023.

**Questions:**

I think the paper is still under-development, it is hard to convince the community with these flow or segmentation evaluations, I would recommend the authors to continue this project, and show more comprehensive evaluations, and solid experiments on these perceptual tasks.

---

### Note · Authors · 2024-11-14

I have read and agree with the venue's withdrawal policy on behalf of myself and my co-authors.